# Anti-Obesity Effects of GABA in C57BL/6J Mice with High-Fat Diet-Induced Obesity and 3T3-L1 Adipocytes

**DOI:** 10.3390/ijms25020995

**Published:** 2024-01-13

**Authors:** Heegu Jin, Hyein Han, Gunju Song, Hyun-Ji Oh, Boo-Yong Lee

**Affiliations:** Department of Food Science and Biotechnology, College of Life Science, CHA University, Seongnam 13488, Republic of Korea; heegu94@hanmail.net (H.J.); hyeinoo@naver.com (H.H.); juhun022188@naver.com (G.S.); guswl264@naver.com (H.-J.O.)

**Keywords:** obesity, energy expenditure, high-fat diet-induced obese mice, 3T3-L1 cells, GABA

## Abstract

Obesity is the excessive accumulation of body fat resulting from impairment in energy balance mechanisms. In this study, we aimed to investigate the mechanism whereby GABA (γ-aminobutyric acid) prevents high-fat diet-induced obesity, and whether it induces lipolysis and browning in white adipose tissue (WAT), using high-fat diet (HFD)-fed obese mice and 3T3-L1 adipocytes. We demonstrated that GABA substantially inhibits the body mass gain of mice by suppressing adipogenesis and lipogenesis. Consistent with this result, histological analysis of WAT demonstrated that GABA decreases adipocyte size. Moreover, we show that GABA administration decreases fasting blood glucose and improves serum lipid profiles and hepatic lipogenesis in HFD-fed obese mice. Furthermore, Western blot and immunofluorescence analyses showed that GABA activates protein kinase A (PKA) signaling pathways that increase lipolysis and promote uncoupling protein 1 (UCP1)-mediated WAT browning. Overall, these results suggest that GABA exerts an anti-obesity effect via the regulation of lipid metabolism.

## 1. Introduction

Obesity is caused by the excessive accumulation of white adipose tissue (WAT), which stores excess energy in the form of triglycerides (TG) [1]. Obesity is a main cause of metabolic syndrome, resulting in disorders such as diabetes, cardiovascular disease, and non-alcoholic fatty liver disease [2]. Therefore, the prevention and treatment of obesity is an active area of research [3]. WAT acts as a crucial regulator of lipid metabolism and energy balance [4].

The differentiation of pre-adipocytes and the subsequent lipid accumulation in mature adipocytes occur through the well-organized processes of hyperplasia and hypertrophy, respectively [5]. During pre-adipocyte differentiation, the expression of CCAAT/enhancer-binding protein alpha (C/EBPα), peroxisome proliferator-activated receptor gamma (PPARγ), and fatty acid-binding protein 4 (FABP4) promote the formation of mature adipocytes and lipid accumulation [6]. In addition, lysophosphatidic acid acyltransferase theta (LPAATθ), lipin 1, fatty acid synthesis (FAS), and diacylglycerol acyltransferase 1 (DGAT1) are involved in the synthesis of triacylglycerol [7].

Lipolysis is the catabolic process that breaks down TGs stored in WAT, resulting in the release of free fatty acids (FFAs) and glycerol [8]. The TG hydrolysis pathway is stimulated by the direct protein phosphorylation of protein kinase A (PKA), which then regulates several key lipolytic proteins that modulate TG lipase activity in adipocytes [9]. The first two primary lipases involved in lipolysis are adipocyte triglyceride lipase (ATGL), which converts TGs to diglycerides, and hormone-sensitive lipase (HSL), which hydrolyses diglycerides to monoglycerides [10]. The final lipase is monoacylglycerol lipase (MGL), which hydrolyzes monoglycerides to liberate further FFAs and glycerol [11]. The released glycerol and FFAs from WAT can be carried in the blood and subsequently infiltrate into other tissues, which distributes lipids and regulates energy balance [12].

Strategies for combating obesity involve not only preventing fat accumulation but also promoting energy expenditure by activating brown adipose tissue (BAT) and the browning of WAT [13]. The expression of uncoupling protein 1 (UCP1) uncouples electron transport in mitochondria, which generates energy as heat [14]. WAT and BAT are readily interconvertible [15]; likewise, the browning of WAT requires the expression of the thermogenic proteins PPARα, PPARγ coactivator 1-alpha (PGC1α), PR domain-containing 16 (PRDM16), and UCP1 [16]. Since these phenotypic changes activate thermogenesis, promoting the browning of WAT may be a therapeutic approach for treating obesity [17].

Current methods for treating obesity include increasing exercise levels, diet control, surgical treatment, and pharmacotherapy [18]. However, these options require time and effort, have high economic costs, and especially in the case of pharmacotherapies, have many side effects, such as rapid weight gain after stopping drug treatment [19]. Therefore, alternative strategies are sought, such as the development of natural anti-obesity agents that are substantially considered safe and without side effects. 

GABA (γ-aminobutyric acid) is a naturally occurring neurotransmitter synthesized from glutamate by the enzyme glutamic acid decarboxylase [20]. Previous studies indicate that GABA is involved in the progression of diabetes by regulating insulin and glucagon levels and glucose homeostasis [21]. However, the potential anti-obesity effect of GABA has not been determined. Therefore, this study aims to characterize the effects of GABA on lipid metabolism and energy expenditure in high-fat diet (HFD)-fed obese mice and 3T3-L1 adipocytes. Our data demonstrate that GABA prevents obesity and related metabolic diseases by decreasing fat accumulation and increasing thermogenesis. 

## 2. Results

### 2.1. GABA Prevents Obesity in HFD-Fed Obese Mice

To determine the anti-obesity effect of GABA in vivo, we fed mice CD, HFD, or HFD supplemented with 10 or 30 mg/kg/day GABA for 8 weeks. At the end of the experimental period, the HFD group was noticeably larger than the CD group, but mice in the GABA10 and GABA30 groups showed a lower body mass than those in the HFD-fed group (Figure 1A). As shown in Figure 1B, the GABA-treated groups had notably less body mass gain as the result of a GABA-induced reduction in WAT accumulation (Figure 1C). Indeed, the mass of the sWAT and vWAT was considerably lower in GABA-treated mice than in HFD-fed mice (Figure 1D), whereas the mass of other tissues, including spleen, lung, and kidney, was not significantly affected (Figure 1E). During the 8 weeks of treatment, water intake was higher in the CD group than in the HFD group, but there was no difference between the HFD, GABA10, and GABA 30 groups. Thus, these results showed that GABA decreased body mass, but food and water intake were not affected by GABA treatment among the diet groups (Figure 1F).

### 2.2. GABA Reduces Hyperglycemia and Hyperlipidemia in HFD-Fed Obese Mice

We next evaluated whether GABA improves HFD-induced glucose intolerance. During the experimental period, fasting blood glucose concentration in the HFD-fed mice increased progressively, whereas the increase was considerably lower in the GABA-treated mice (Figure 2A). As shown in Figure 2B, the GABA-treated mice showed improved glucose tolerance in the 2 h OGTT. GABA-treated mice had a more rapid decline in blood glucose concentration than HFD-fed mice. At the end of the 8 weeks of GABA treatment, we evaluated the serum lipid profiles. The results showed that GABA reduced HFD-induced disorders in the serum lipid profile, such as insulin, TG, total cholesterol, LDL cholesterol, and HDL cholesterol concentrations (Figure 2C–G). These results show that GABA ameliorates hyperglycemia and dyslipidemia in HFD-fed obese mice.

### 2.3. GABA Regulates Lipid Metabolism by Suppressing Adipogenesis and Lipogenesis

The reduction in body mass, and improvement in glucose tolerance and dyslipidemia induced by GABA treatment, were accompanied by a decrease in the size of WAT. To examine the effect of GABA on WAT mass, fat and lean mass were measured using DEXA (Figure 3A). The fat percentage of the HFD group was higher than those of the CD group, and GABA treatment dose-dependently lowered the fat percentage (Figure 3B). As shown in Figure 3C, H&E staining showed that adipocyte size in HFD-fed obese mice was much larger than that in CD-fed obese mice. However, the adipocytes of GABA-treated mice were smaller with less lipid accumulation (Figure 3D). In addition, we performed Western blot analysis to determine the molecular mechanisms of the effects of GABA on lipid accumulation. The protein expression levels of adipogenic factors (C/EBPα, PPARγ, and FABP4) and lipogenic factors (LPAATθ, lipin1, DGAT1, and SREBP1) in both sWAT and vWAT were considerably higher in HFD-fed mice than in CD-fed mice; this finding suggests that levels of specific proteins in adipocytes can be used to characterize an obese state that is associated with hypertrophy and hyperplasia. However, the expression levels of the adipogenesis and lipogenesis factors were much lower in mice that were fed GABA (Figure 3E,F). These results suggest that GABA reduces lipid accumulation in fat depots by inhibiting adipogenesis and lipogenesis.

### 2.4. GABA Promotes Energy Expenditure by Increasing Lipolysis and Browning in WAT

Increased lipolysis and UCP1 expression levels are hallmarks of WAT browning [16]. To determine whether GABA stimulates lipolysis, we measured PKA phosphorylation and the expression levels of lipolytic proteins, including ATGL, p-HSL, and MGL, through Western blot analysis. Expression of PKA was significantly increased in GABA-treated mice; this finding indicates that GABA increases lipolysis in WAT by activating PKA phosphorylation and upregulating the expression of lipolytic enzymes (Figure 4A,B). In addition, the rectal temperature of the HFD group was lower than that of the CD group, whereas the rectal temperature of the GABA-treated group was significantly higher than that of the non-treated groups (Figure 4C,D). Consequently, we next measured the expression levels of proteins that induce thermogenesis. As shown in Figure 4A,B, GABA treatment induced the expression of browning proteins encoding PPARα, PGC1α, PRDM16, and UCP1. In accordance with the above findings, the immunofluorescence intensity of PKA and UCP1 was increased in GABA-treated mice (Figure 5A,B). Taken together, these results indicate that GABA promotes lipolysis and browning, therefore resulting in a greatly increased energy expenditure as heat.

### 2.5. GABA Suppresses Hepatic Lipid Accumulation in the Liver

When adipose tissue is unable to store excess fat, lipids accumulate inappropriately in the liver, inducing fatty liver disease in HFD-fed obese mice [22]. Thus, we established whether GABA decreased hepatic lipid accumulation in HFD-fed obese mice. As shown in Figure 6A, the color of HFD-fed obese livers was yellowish-brown, whereas the livers of the GABA-treated group exhibited a normal reddish-brown, almost likely that of the CD group. Moreover, HFD-fed obese liver ORO staining showed significant hepatic fat content (Figure 6B). However, GABA significantly decreased hepatic lipid accumulation and liver mass in HFD-fed mice (Figure 6C). We also found that GABA reduced serum AST and ALT activity, which are biochemical features of liver disease (Figure 6D,E). In addition, the expression levels of hepatic lipogenic proteins (LPAATθ, lipin1, DGAT1, SREBP1, and FAS) in the liver were detected by Western blot analysis. As shown in Figure 6F, HFD-fed obese mice expressed high levels of these proteins, but GABA decreased hepatic lipid accumulation through the downregulation of lipogenesis in the liver.

### 2.6. GABA Reduces Lipid Accumulation in 3T3-L1 Adipocytes

Next, the anti-obesity activity of GABA was evaluated in vitro in a 3T3-L1 cell model. Before evaluating the effects of GABA, we first assessed the cytotoxicity of GABA in 3T3-L1 cells. As shown in Figure 7A, 40 μg/mL GABA was cytotoxic; therefore, concentrations of 2.5, 5, 10, or 20 μg/mL GABA were added to 3T3-L1 cell medium to determine how GABA affects 3T3-L1 adipocyte differentiation and lipid accumulation. As shown in Figure 7B–D, GABA prevented adipocyte differentiation and decreased intracellular lipid droplets. These data indicate that GABA reduces lipid accumulation during 3T3-L1 cell differentiation.

### 2.7. GABA Regulates Lipid Metabolism in 3T3-L1 Adipocytes

To determine the molecular mechanisms by which GABA affects lipid metabolism in vitro, we performed Western blot analysis in 3T3-L1 cells. Consistent with the in vivo results, GABA decreased the expression levels of adipogenic factors in a concentration-dependent manner (Figure 8A) and significantly decreased the expression levels of lipogenic markers in 3T3-L1 adipocytes (Figure 8B). We next determined whether GABA regulates the expression of lipolytic enzymes and proteins involved in WAT browning in 3T3-L1 adipocytes. As expected, dose-dependent significant upregulation in the expression levels of lipolytic and browning proteins was shown in GABA-treated 3T3-L1 adipocytes (Figure 8C,D). Taken together, these results show that GABA decreases the expression levels of proteins involved in adipogenesis and lipogenesis and increases lipolysis and thermogenesis in 3T3-L1 adipocytes involved in the anti-obesity effect.

## 3. Discussion

Obesity, and its many associated diseases such as diabetes, hypertension, osteoarthritis, and heart disease, has become a major worldwide health problem. Obesity not only reduces quality of life but also shortens life expectancy and burdens society with economic and social problems [23]. To find a sustainable way to prevent obesity, this study investigated the anti-obesity activity of GABA in suppressing fat accumulation and WAT browning-induced energy expenditure.

GABA is an amino acid that acts as an inhibitory neurotransmitter in the central nervous system. It is widely distributed in cortical neurons and decreases neuronal excitability in the cortex by reducing neuronal action potentials [24]. Several reports have shown that GABA administration improves glucose intolerance and insulin sensitivity, and has anti-inflammatory and antioxidant effects and other beneficial properties in animal models [25]. However, it has not been established whether GABA has an anti-obesity effect. To evaluate the potential anti-obesity effect of GABA, we employed a HFD-induced obesity mouse model and 3T3-L1 adipocytes. HFDs are widely used to induce an overweight phenotype and fat deposition in animals. The 3T3-L1 adipocytes are also a well-established in vitro model. Current research has shown that GABA not only decreases lipid accumulation, but also promotes energy expenditure, resulting in an improved overall metabolic status, as indicated by improvements in glucose tolerance, hyperglycemia, and hepatic steatosis.

HFD-induced obesity is related to excessive fat accumulation [26]. As expected, HFD-fed mice over a period of 8 weeks became obese and exhibited significant increases in the mass of sWAT and vWAT depots. However, GABA dose-dependently reduced body and WAT mass, improved fasting blood glucose concentrations, and reduced the size of WAT depots in these mice. In addition, there were no significant differences in fecal excretion or nutrition absorption among the diet groups. Therefore, food and water intake, fecal excretion, and nutrition absorption were not affected by GABA treatment. The mass of WAT can be regulated by adipocyte hyperplasia or hypertrophy, so we measured the expression of adipogenic proteins that are essential in pre-adipocyte differentiation, a factor that contributes to hyperplasia. Moreover, TG synthesis causes adipocyte hypertrophy, which requires the expression of lipogenic proteins [27]. GABA decreased the HFD-induced expression of adipogenic and lipogenic proteins. This finding indicates that GABA reduces HFD-induced body mass gain by reducing TG accumulation and WAT expansion. In addition, GABA decreases fasting blood glucose concentrations and thus improves glucose tolerance in HFD-fed obese mice.

Obesity occurs when energy intake is greater than energy expenditure; therefore, promoting energy expenditure is an emerging approach to preventing obesity [27]. TG hydrolysis delivers fuel for fatty acid oxidation, and this pathway is stimulated by lipolytic enzymes including p-PKA, ATGL, HSL, and MGL [28]. In the present study, we found that GABA upregulated the lipolysis pathway in HFD-fed obese mice. Therefore, GABA may induce energy consumption by increasing the expression of ligases in WAT. Likewise, WAT browning—the stimulation of the brown adipocyte-like phenotype in WAT—is another way to stimulate energy consumption [29]. BAT dissipates energy as heat during thermogenesis and stimulating this process offers a new way to combat obesity and obesity-related diseases [30]. To establish whether GABA induces thermogenesis, we measured the rectal temperature of the mice and found that HFD significantly decreased rectal temperature, but the GABA-treated group had a higher rectal temperature than that of the other groups. This finding suggests that the GABA-treated group maintained thermogenic capacity even when consuming a HFD. We also determined the browning phenotype of WAT and found the expression levels of browning proteins in GABA-treated mice. These results suggest that GABA increased the expression of thermogenic genes in WAT from HFD-fed mice. Moreover, to determine whether GABA activates PKA and thus promotes UCP1 activity, we performed immunofluorescence double staining. Consistent with the results described above, the immunoreactivity of PKA and UCP1 was increased in adipose tissue from GABA-treated mice. Immunostaining data of adipose tissue show that GABA treatment dose-dependently controls the PKA-mediated lipolysis pathway and ultimately induces UCP1. Additionally, UCP1, in the inner membrane of mitochondria, uncouples fuel oxidation from ATP synthesis. These results suggest that GABA promotes energy expenditure through WAT browning in HFD-fed obese mice. 

A HFD increases fat accumulation, not only in WAT but also in the liver, resulting in hepatic steatosis and non-alcoholic fatty liver disease in mouse models and humans [31]. In this study, we found that a HFD caused ectopic fat deposition in the liver, as evidenced by its yellow color and increased weight, and this effect was prevented by GABA treatment. In addition, lipid accumulation in the liver is initiated by activating the hepatic lipogenesis pathway [32], whose expression was decreased by GABA in this study. Therefore, our results indicate that GABA suppresses hepatic lipid accumulation in HFD-fed obese mice. 

We also studied the effect of GABA using the 3T3-L1 cell-line model. Adipogenesis is regulated by a network of multiple differentiation genes, which induces the process and promotes the expression of downstream FFA synthesis genes [33]. ORO staining showed that GABA reduced lipid accumulation in a dose-dependent manner. In addition, Western blot analysis confirmed that GABA inhibited lipid droplet accumulation by decreasing the protein levels of adipogenesis and lipogenesis factors. Moreover, GABA upregulated the expression of lipolytic enzymes and thermogenic genes; these results are consistent with the in vivo data in this study. Together, these results confirm, in vivo and in vitro, that GABA has important effects on lipid metabolism.

## 4. Materials and Methods

### 4.1. Preparation of GABA

GABA (A5835; Sigma-Aldrich, St. Louis, MO, USA) used in this study was dissolved in distilled water and administered orally to mice at a dose of 10 or 30 mg/kg daily.

### 4.2. Animals and Treatments

The animal studies were performed according to the criteria outlined in the “Guide for the Care and Use of Laboratory Animals” authored by the US National Academy of Sciences and published by the US National Institutes of Health and were approved by the Institutional Animal Care and Use Committee of CHA University (IACUC, Approval Number 220145). Male C57BL/6J mice, 8 weeks old, were purchased from Raon Bio (Yongin, Republic of Korea) and housed in a temperature- and humidity-regulated facility under a 12 h light/dark cycle. After 1 week of adaptation, the mice were randomly allocated to four groups (*n* = 15 per group) as follows: a chow diet (CD) group, a high-fat diet (HFD) group, a group fed HFD supplemented with oral GABA at 10 mg/kg/day (GABA10), or a group fed HFD supplemented at 30 mg/kg/day (GABA30) for 8 weeks. The HFD contained 60 kcal% as fat (D12492, Research Diets, New Brunswick, NJ, USA) and the CD contained 10 kcal% as fat (D12450B, Research Diets). At the end of the experiment, mice were fasted for 12 h, euthanized using CO_2_, and then blood and tissue samples were collected.

### 4.3. Body Mass and Dietary Intake Measurements

The body masses of the mice were measured weekly using an analytical balance. Food and water intake were calculated each week as the differences between the initial quantities supplied and the amounts of food and water remaining after 1 week.

### 4.4. Fasting Blood Glucose Measurement

The fasting glucose concentrations were measured in blood samples collected weekly from a tail vein after 12 h of fasting using an Accu-Check blood glucose meter (Roche, Basel, Switzerland).

### 4.5. Oral Glucose Tolerance Testing

Oral glucose tolerance testing (OGTT) was performed after overnight fasting. The mice were administered 1.5 g/kg body mass D-glucose orally, and then the glucose concentration in blood samples collected from a tail vein was measured after 0, 30, 60, 90, and 120 min using an Accu-Check blood glucose meter.

### 4.6. Rectal Temperature Measurement

The rectal temperatures of the mice were measured weekly using a Testo 925 thermometer (Testo, Lenzkirch, Germany).

### 4.7. Dual-Energy X-ray Absorptiometry

Mouse body composition, including fat and lean mass, was analyzed using dual-energy X-ray absorptiometry (DEXA) with an InAlyzer dual X-ray absorptiometer (Medikors, Seongnam, Republic of Korea).

### 4.8. Biochemical Analysis

Blood samples were obtained by cardiac puncture under terminal anesthesia and centrifuged at 3000× *g* for 20 min at 4 °C to collect serum. The serum concentrations of insulin were measured using a mouse metabolic hormone magnetic bead panel (Merck Millipore, Burlington, MA, USA). The serum concentrations of insulin, TG, total cholesterol, low-density lipoprotein (LDL) cholesterol, and high-density lipoprotein (HDL) cholesterol, and the activities of aspartate aminotransferase (AST) and alanine aminotransferase (ALT), were determined using colorimetric assay kits (Roche).

### 4.9. Histological Analysis

Subcutaneous (sWAT) and visceral (vWAT) WAT samples were fixed in 4% paraformaldehyde and embedded in paraffin. Multiple sections were then prepared and stained with hematoxylin and eosin (H&E) for histological evaluation. Photomicrographs were obtained using a Nikon E600 microscope (Nikon, Tokyo, Japan).

### 4.10. Immunofluorescence

WAT sections were deparaffinized and then incubated with anti-PKA or anti-UCP1 antibodies. Secondary anti-mouse fluorescein isothiocyanate (FITC)-conjugated and anti-rabbit Alexa Fluor 594-conjugated antibodies were then applied. DAPI (Thermo Fisher Scientific, Waltham, MA, USA) was used to stain the cell nuclei and the sections were then mounted using ProLong Gold Antifade reagent (Thermo Fisher Scientific). Fluorescent images were captured using a Zeiss confocal laser scanning microscope (LSM880; Carl Zeiss, Oberkochen, Germany) and Zen 2012 software (Version 1.1.1.0, Carl Zeiss).

### 4.11. Cell Culture

3T3-L1 pre-adipocytes were purchased from the American Type Culture Collection (Manassas, VA, USA) and cultured in Dulbecco’s modified Eagle’s medium (DMEM) containing 10% bovine calf serum (BS) and 1% penicillin/streptomycin (P/S) until confluent. After 2 days (D0), this medium was replaced with DMEM containing 10% fetal bovine serum (FBS), 1% P/S, and MDI (0.5 mM 3-isobutyl-1-methylxanthine, 1 μM dexamethasone, and 4 μg/mL insulin). On D2, this medium was replaced with DMEM containing 10% FBS, 1% P/S, and 4 μg/mL insulin, and this medium was replaced every 2 days until day 8 (D8). For GABA treatment, 2-day confluent 3T3-L1 cells were incubated with different concentrations of GABA (0, 2.5, 5, 10, or 20 μg/mL) every 2 days during differentiation until mature adipocytes had formed (D8).

### 4.12. Cell Viability

To determine the appropriate concentrations of GABA to be used in further experiments, a cell viability assay was performed using 3-(4,5-dimethylthiazol-2-yl)-2,5-diphenyltetrazolium bromide (MTT, Thermo Fisher Scientific). 3T3-L1 pre-adipocytes were seeded into 96-well plates and incubated overnight. Stock solutions of GABA were prepared in DMSO and then the cells were treated with various concentrations of GABA (0, 2.5, 5, 10, 20, or 40 μg/mL) for 24 h. The MTT solution was added to each well, and the cells were incubated for a further 3 h. After removing the MTT-containing medium, DMSO was added to elute the formazan crystals. The absorbances of these eluates were then measured at 570 nm using a Biotek ELISA reader (BioTek, Winooski, VT, USA).

### 4.13. Oil Red O Staining

After 3T3-L1 cells were differentiated for 8 days, the fully differentiated adipocytes were fixed in 4% formaldehyde for 1 h and then washed twice with 60% isopropanol. The fixed cells were stained with Oil Red O (ORO) solution for 30 min and then washed with distilled water. After drying, the stained cells were imaged, and the stain was eluted using 100% isopropanol. To analyze hepatic lipid accumulation, cryostat sections of the liver were stained with ORO solution and photomicrographs were obtained using a Nikon E600 microscope.

### 4.14. Western Blot Analysis

Tissues were washed twice with PBS and then lysed in lysis buffer (1 mM phenylmethylsulfonyl fluoride, 1 mM ethylenediaminetetraacetic acid, 1 μM pepstatin A, 1 μM leupeptin, and 0.1 μM aprotinin; iNtRON Biotechnology, Seoul, Republic of Korea) containing phosphatase and protease inhibitors, and incubated on ice for 1 h to permit lysis. After homogenization and centrifugation at 3000× *g* for 20 min at 4 °C, the protein content in the supernatant was determined, and the lysate protein concentrations were quantified using a protein assay kit (Bio-Rad, Hercules, CA, USA). Lysates containing equal amounts of protein were separated using SDS-PAGE and the proteins were electro-transferred to membranes. Then, the membranes were blocked using 5% skim milk for 1 h, washed with Tris-buffered saline containing Tween 20 (TBST), incubated with primary antibodies overnight at 4 °C, and then exposed to horseradish peroxidase-conjugated secondary antibodies. Antibodies targeting C/EBPα, PPARγ, FABP4, sterol regulatory element-binding protein 1 (SREBP1), LPAATθ, lipin 1, DGAT1, phosphorylated PKA (p-PKA, Ser 114), PGC1α, and glyceraldehyde 3-phosphate dehydrogenase (GAPDH) were purchased from Santa Cruz Biotechnology (Dallas, TX, USA); antibodies targeting ATGL and phosphorylated HSL (p-HSL, Ser 563) were purchased from Cell Signaling Technology (Danvers, MA, USA); antibodies targeting MGL, PPARα, PRDM16, and UCP1 were purchased from Abcam (Cambridge, UK).

### 4.15. Statistical Analysis

Data are expressed as the mean ± SEM. Statistical comparisons were made using one-way ANOVA followed by Tukey’s post-hoc test (IBM SPSS Statistics Version 20.0, Armonk, NY, USA). *p* < 0.05 was regarded as indicating statistical significance.

## 5. Conclusions

In conclusion, we found that GABA suppressed the HFD-mediated increases in body mass and fat accumulation by reducing adipogenesis and lipogenesis. Moreover, GABA promoted WAT browning, increasing the loss of energy as heat. GABA improved hyperglycemia, dyslipidemia, and hepatic steatosis in HFD-fed mice. Overall, these findings suggest that the efficacy and safety of GABA warrant that it be investigated further as a therapeutic agent for the treatment of obesity and its associated metabolic disorders.

## Figures and Tables

**Figure 1 ijms-25-00995-f001:**
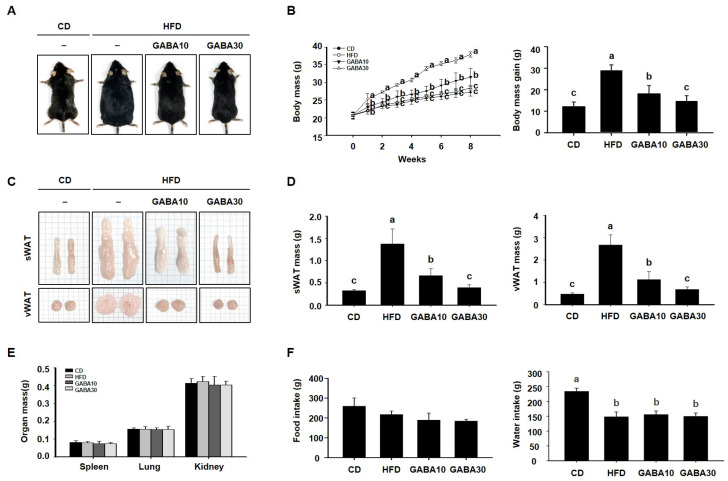
The effects of GABA on mice fed a HFD. (**A**) Representative images of the mice. (**B**) Weekly body mass during 8 weeks of GABA treatment. (**C**) Representative images of the sWAT and vWAT. (**D**) sWAT and vWAT mass after 8 weeks of GABA treatment. (**E**) The mass of other organs. (**F**) Food and water intake during the 8 weeks of GABA treatment. Data are expressed as the mean ± SEM. Values indicated by different letters are significantly different; *p* < 0.05 (a > b > c).

**Figure 2 ijms-25-00995-f002:**
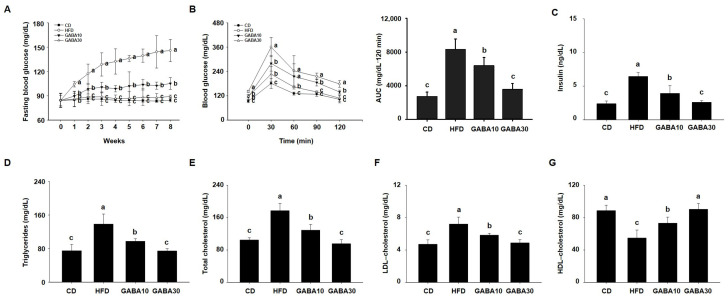
The effects of GABA on hyperglycemia and dyslipidemia in HFD-fed obese mice. (**A**) The fasting blood glucose, (**B**) oral glucose tolerance, (**C**) serum insulin, (**D**) TG, (**E**) total cholesterol, (**F**) LDL cholesterol, and (**G**) HDL cholesterol concentrations after 8 weeks of GABA treatment. Data are expressed as the mean ± SEM. Values indicated by different letters are significantly different; *p* < 0.05 (a > b > c).

**Figure 3 ijms-25-00995-f003:**
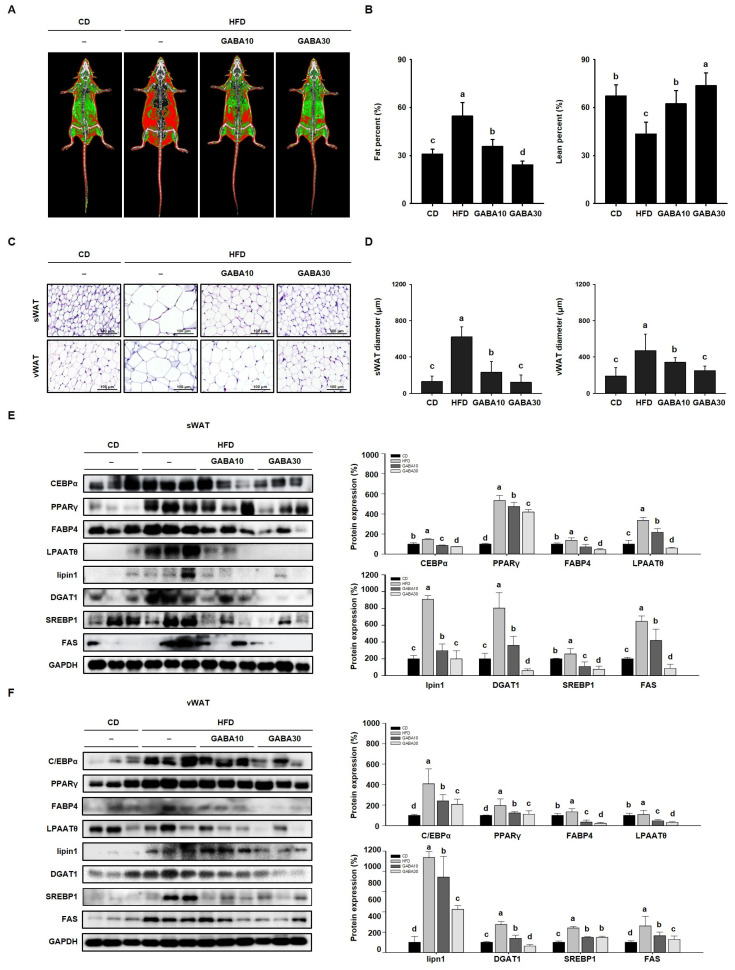
The effects of GABA on lipid accumulation in HFD-fed obese mice. (**A**) Body composition images obtained using DEXA. Fat tissue is shown in red and lean is in green. (**B**) Fat and lean percentage. (**C**) H&E staining of sWAT and vWAT. Scale bar: 100 μm. (**D**) sWAT and vWAT diameter. (**E**) Western blots of adipogenic proteins (C/EBPα, PPARγ, and FABP4) and lipogenic proteins (LPAATθ, lipin1, DGAT1, SREBP1, and FAS) in sWAT and (**F**) vWAT. Data are expressed as the mean ± SEM. Values indicated by different letters are significantly different; *p* < 0.05 (a > b > c > d).

**Figure 4 ijms-25-00995-f004:**
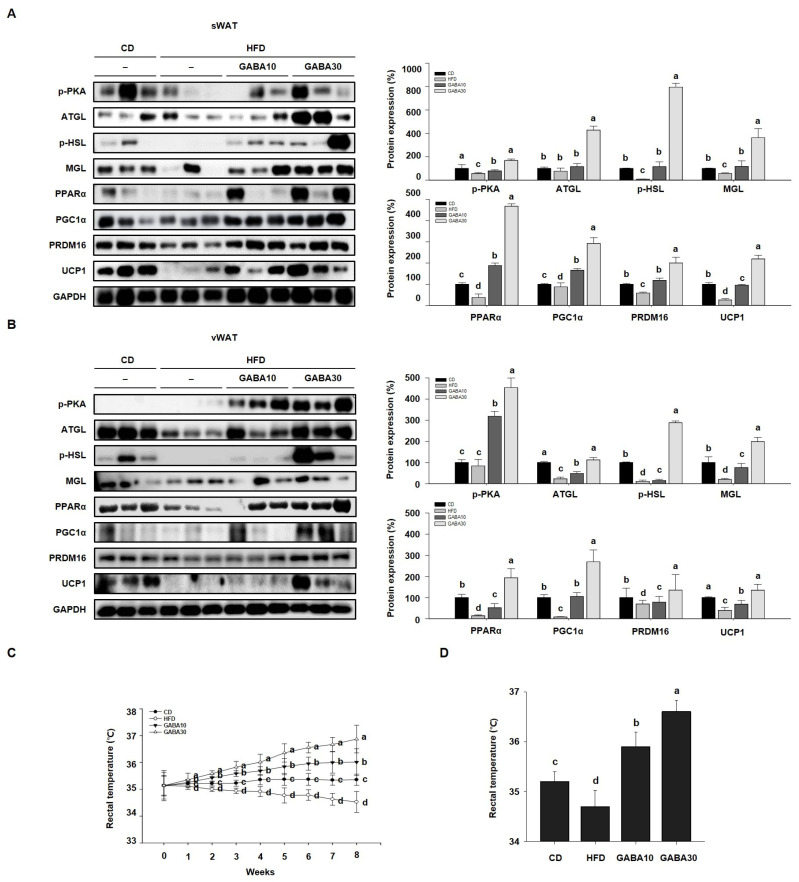
The effects of GABA on energy expenditure in HFD-fed obese mice. (**A**) Western blots of lipolytic enzymes (p-PKA, ATGL, p-HSL, and MGL) and browning proteins (PPARα, PGC1α, PRDM16, and UCP1) in sWAT and (**B**) vWAT. (**C**) Rectal temperature during the 8-week experimental period. (**D**) Rectal temperature after 8 weeks of GABA treatment. Data are expressed as the mean ± SEM. Values indicated by different letters are significantly different; *p* < 0.05 (a > b > c > d).

**Figure 5 ijms-25-00995-f005:**
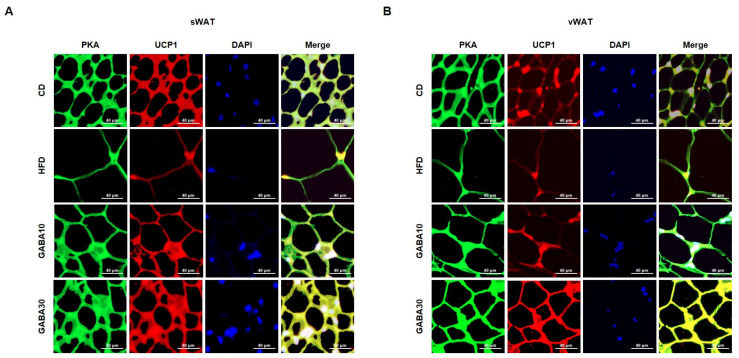
Double immunofluorescence staining of WAT to determine lipolysis and browning. (**A**) Immunofluorescence images of the expression of PKA (green), UCP1 (red), and nuclei (blue) in sWAT and (**B**) vWAT. The yellowish color in the Merge image indicates the overlapping of the green and red colors. Scale bar: 40 μm.

**Figure 6 ijms-25-00995-f006:**
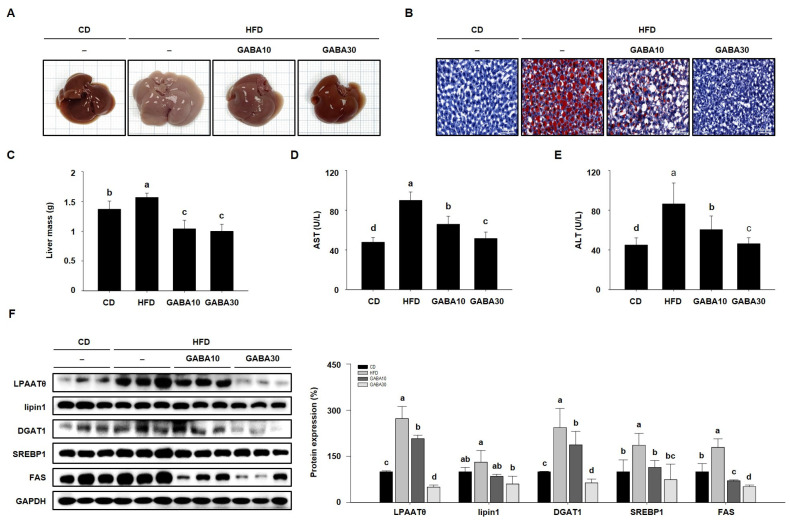
The effects of GABA on hepatic lipid accumulation in HFD-fed obese mice. (**A**) Representative images of livers. (**B**) ORO staining of liver is shown with lipid droplets in red. Scale bar: 50 μm. (**C**) Liver mass, (**D**) serum AST, and (**E**) ALT activities after 8 weeks of GABA treatment. (**F**) Western blots of hepatic lipogenic proteins (LPAATθ, lipin1, DGAT1, SREBP1, and FAS). Data are expressed as the mean ± SEM. Values indicated by different letters are significantly different *p* < 0.05 (a > b > c > d).

**Figure 7 ijms-25-00995-f007:**
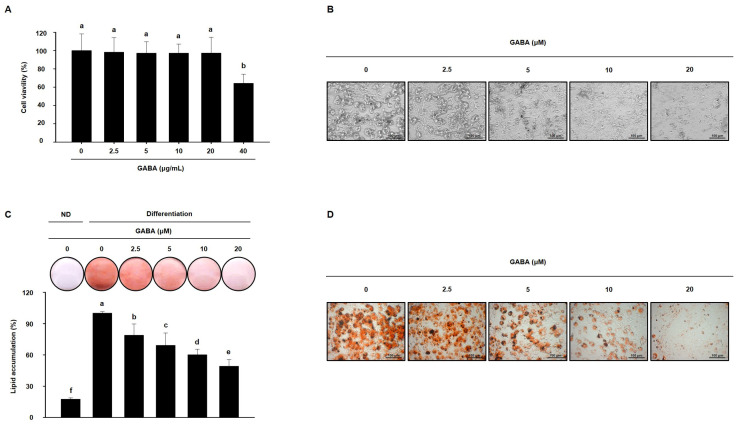
The effects of GABA on lipid accumulation in 3T3-L1 adipocytes. (**A**) Viability of 3T3-L1 adipocytes treated with GABA for 24 h. (**B**) Microscopic images of 3T3-L1 adipocytes after 8 days of differentiation. Scale bar: 100 μm. (**C**) ORO staining and (**D**) Microscopic images of ORO-stained fully differentiated 3T3-L1 adipocytes are shown with lipid droplets in red. Scale bar: 100 μm. Data are expressed as the mean ± SEM. Values indicated by different letters are significantly different *p* < 0.05 (a > b > c > d > e > f).

**Figure 8 ijms-25-00995-f008:**
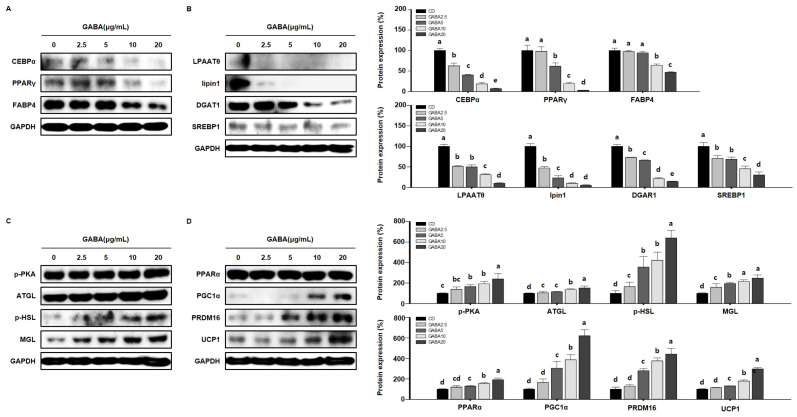
The effects of GABA on lipid regulation in differentiated 3T3-L1 adipocytes. (**A**) Western blots of adipogenic proteins (C/EBPα, PPARγ, and FABP4), (**B**) lipogenic proteins (LPAATθ, lipin1, DGAT1, and SREBP1), (**C**) lipolytic enzymes (p-PKA, ATGL, p-HSL, and MGL), and (**D**) browning proteins (PPARα, PGC1α, PRDM16, and UCP1) in 3T3-L1 adipocytes. Data are expressed as the mean ± SEM. Values indicated by different letters are significantly different *p* < 0.05 (a > b > c > d > e).

## Data Availability

All data generated or analyzed during this study are included in this published article.

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
