# Peer review of "Anti-Obesity Effects of GABA in C57BL/6J Mice with High-Fat Diet-Induced Obesity and 3T3-L1 Adipocytes"

_ijms, 2024, doi:10.3390/ijms25020995_

Round 1
Reviewer 1 Report
Comments and Suggestions for Authors
This paper investigates the influence of GABA on weight gain, fat metabolism, and a plethora of biochemical markers in mice fed a both normal and high fat diets.
The study convincingly suggests that GABA co-administered with a high fat diet seems to maintain the mice in a normal metabolic state unlike the mice fed the high fat diet.
The paper shows a logical flow and the results are well described. I have no recommendations for improvement.
It would be interesting to see if mice that are already obese could lose weight if administered GABA, although this is probably implied by the research findings.
Author Response
Response to Reviewer 1 Comments
We appreciate you for your time and effort in consideration of our manuscript. Your kind advice will help us set up the research that should be directed in the future. Again, thank you for your valuable suggestion.
Reviewer 2 Report
Comments and Suggestions for Authors
In this paper, the authors aimed to investigate the mechanism whereby GABA (γ-aminobutyric acid) prevents high-fat diet-induced obesity and whether it induces lipolysis and browning in white adipose tissue (WAT), using high-fat diet (HFD)-fed obese mice and 3T3-L1 adipocytes. However the authors choose GAPDH as the reference protein in the study. As it has been reported that GAPDH expression changes with high fat diet feeding, therefore it is not suitable to be used as a reference protein.
Here are some of the concerns with the paper:
Page 1 line 25: “Obesity is caused by the accumulation of white adipose tissue (WAT),”
Only excessive accumulation can cause obesity, normal accumulation of WAT is healthy storage of lipids in WAT.
Line 36: “lipid production”, it should be lipid accumulation here.
For lipid production, the involvement of enzymes such as fatty acid synthase and DGAT should be mentioned.
Figure 1, A body composition analysis, such as DXA, should be performed to assess the changes in % of lean mass and fat mass with different treatment.
The measurement of vWAT and sWAT only provide part of the picture of the changes that accompanies the changes of body mass. What is the change of mass of brown adipose tissue?
Figure 3, It has been reported that GAPDH level changes with high fat feeding, therefore, GAPDH should not be used as internal control for studies involving high fat diet feeding. Please choose another internal control for this experiment. Front Nutr. 2020; 7: 589771. PMCID: PMC7732482; PMID: 33330591
For the quantification of the bands in the blots, some of the protein show very different levels in the lanes of the same treatment, ie. LPAAT8, Lipin1 DGAT1 and FAS, however, there seem to be not much of a difference when shown in the quantification (the error bar seems to be very small). What software was used in the quantification? Can you show a table with the digital conversion of the bands?
Figure 4: Same issue with the choice of GAPDH as reference protein. Please choose another internal control for the study.
Figure 5,The total TG and TC in the liver should be measured to document the changes.
A metabolic cage or indirect calorimetry assay of energy expenditure should be performed on these mice with different diet treatment.
Line 289-292. Why do both the HFD and CD have the same cat. #? What are the diets used for this study?
Author Response
Response to Reviewer 2 Comments
"Please see the attachment."
We thank you for your time and effort in giving us the opportunity to strengthen our manuscript with your valuable comments. Thus, it is with great pleasure that we resubmit our manuscript for further consideration.
To facilitate your review of our revisions, the following is a point-by-point response to the questions and comments: the original reviewer comments are provided in black color, whereas our answers are given in red. The appropriate changes made in the revised manuscript are highlighted in Microsoft Word.
Again, thank you for giving us the opportunity to strengthen our manuscript with your valuable comments. We have worked hard to incorporate your feedback and hope that the revised manuscript is suitable for publication in International Journal of Molecular Sciences.

Round 2
Reviewer 2 Report
Comments and Suggestions for Authors
This paper has its own unique findings and there is improvement in the revised manuscript, however, the authors should be careful about how the draw their conclusions.
1) Without the proper measurement of energy expenditure using a metabolic cage or indirect calorimetry assay of energy expenditure, the authors should not make the statement about higher energy expenditure with the treatment (such as line 144). Higher rectal temperature can be suggestive about a higher energy expenditure, but not a proof.
The food and water intake was documented in figure 1, how about the amount of feces excreted from the animals? Were there difference in the fecal excretion with the treatment? did the animals have difference in nutrition absorption?
2) Brown adipose tissue has significant contribution to whole body energy expenditure, yet the authors did not measure the brown adipose tissue in this paper. There would be changes in brown adipose tissue with high fat feeding, what happens in terms of brown adipose treatment with GABA treatment?
3) The in vitro data shows that GABA treatment decreased lipid accumulation in differentiated 3T3-L1 adipocytes. Together with the in vivo data showing decreased WAT adipose tissues with GABA treatment as well as pattern of protein expression indicating an increase in lipolysis. However there is no functional assay for lipolysis, which can be performed in vitro (using WAT tissue) or in vivo (measuring fatty acid and glycerol release in the serum). If analysis of energy expenditure is off limit for the lab, the authors should at least consider some functional assays for lipolysis to proof what the indications suggested by the protein levels are actually holds true. Without functional assays for lipolysis and energy expenditure, all the data presented with western blot can only be considered as suggestive.
The authors should consider change title of the paper to reflect what they studied and what solid evidence do they have. The authors should also address the limitations in their study design and discuss their results in the context of the result/evidence they have on the subject.
Author Response
Response to Reviewer 2 Comments (Round 2)
"Please see the attachment."
We thank you for your time and effort in giving us the opportunity to strengthen our manuscript with your valuable comments. Thus, it is with great pleasure that we resubmit our manuscript for further consideration.
To facilitate your review of our revisions, the following is a point-by-point response to the questions and comments: the original reviewer comments are provided in black color, whereas our answers are given in red. The appropriate changes made in the revised manuscript are highlighted in Microsoft Word.
Again, thank you for giving us the opportunity to strengthen our manuscript with your valuable comments. We have worked hard to incorporate your feedback and hope that the revised manuscript is suitable for publication in International Journal of Molecular Sciences.

Round 3
Reviewer 2 Report
Comments and Suggestions for Authors
In your response 1:
Moreover, GABA reduced body mass, but there were no significant differences in the fecal excretion or in nutrition absorption among the diet groups. Therefore, food and water intake, fecal excretion, and nutrition absorption were not affected by GABA treatment.
Please show data to support this comment; fecal excretion data and nutrition absorption data. If there is no data to show, these can be mentioned in the discussion.
For metabolic studies, it is understandable that not all the aspect is included in one paper, therefore it is important in the result to stick to what observation was made and what data was collected so that to not confuse the readers of facts and hypothesis. Discussion is more of the place to express authors' thinking.
Author Response
Response to Reviewer 2 Comments
"Please see the attachment."
We thank you for your time and effort in giving us the opportunity to strengthen our manuscript with your valuable comments. Thus, it is with great pleasure that we resubmit our manuscript for further consideration.
To facilitate your review of our revisions, the following is a point-by-point response to the questions and comments: the original reviewer comments are provided in black color, whereas our answers are given in red. The appropriate changes made in the revised manuscript are highlighted in Microsoft Word.
Again, thank you for giving us the opportunity to strengthen our manuscript with your valuable comments. We have worked hard to incorporate your feedback and hope that the revised manuscript is suitable for publication in International Journal of Molecular Sciences.
Pont 1: In your response 1:
Moreover, GABA reduced body mass, but there were no significant differences in the fecal excretion or in nutrition absorption among the diet groups. Therefore, food and water intake, fecal excretion, and nutrition absorption were not affected by GABA treatment.
Please show data to support this comment; fecal excretion data and nutrition absorption data. If there is no data to show, these can be mentioned in the discussion.
For metabolic studies, it is understandable that not all the aspect is included in one paper, therefore it is important in the result to stick to what observation was made and what data was collected so that to not confuse the readers of facts and hypothesis. Discussion is more of the place to express authors' thinking.
Response 1: Thank you for your valuable suggestion. As mentioned in revised version 3, there were no significant differences in the fecal excretion or in nutrition absorption among the diet groups, but only dietary intake data were collected to show. Instead, there are no fecal excretion and nutrition absorption data to show, therefore as your kind advice, we have mentioned that we observed in the Discussion section. We hope that the revised statement clarifies what observation was made and what data was collected as to not confuse the readers of facts and hypotheses.
Revised line 239-242: Also, there were no significant differences in the fecal excretion or in nutrition absorp-tion among the diet groups (data not shown). Therefore, food and water intake, fecal excretion, and nutrition absorption were not affected by GABA treatment.
